# Fabrication of a Hot-Embossing Metal Micro-Mold through Laser Shock Imprinting

**DOI:** 10.3390/ma16145079

**Published:** 2023-07-19

**Authors:** Haifeng Yang, Jingbin Hao, Haoda Wang, Mengsen Ding

**Affiliations:** 1School of Mechatronic Engineering, China University of Mining and Technology, Xuzhou 221116, China; yhf002@163.com (H.Y.); wanghaoda@cumt.edu.cn (H.W.); ms_ding@cumt.edu.cn (M.D.); 2Jiangsu Key Laboratory of Mine Mechanical and Electrical Equipment, China University of Mining & Technology, Xuzhou 221116, China

**Keywords:** laser shock imprinting, metal micro-mold, hot embossing

## Abstract

As a technology for polymer surface fabrication, roll-to-roll hot embossing has been widely used because of its advantages, such as its low cost and high efficiency. However, the metal micro-mold is a major factor that determines the hot embossing of the polymer. In this study, a new metal micro-mold manufacturing method is proposed. The metal micro-mold is produced using laser shock imprinting (LSI) on the surface of metal foil. It has the characteristics of good thermal stability and high strength. During our LSI experiment, the strength of the mold increased after laser shocking. In this study, copper foils of different thicknesses were selected for LSI experiments. Through the analysis of the profile and forming depth of the microstructure, combined with the numerical simulation of the forming mechanism of copper foils with different thicknesses using ABAQUS software(Abaqus 2021), a copper foil with a flat back was selected as the final metal micro-mold. On this basis, copper molds with different microstructure shapes were created. Then, the mold was used in the hot-embossing experiment to manufacture the microstructure on the surface of polyethylene terephthalate (PET) and to study the fidelity and integrity of the molded microstructure. The deformation resistance of the copper mold under hot-embossing conditions was verified through a nano-indentation experiment. The final results show that the metal micro-mold produced via LSI had a high accuracy and molding stability and has potential applications in the field of roll-to-roll hot embossing.

## 1. Introduction

Chou et al. [1] of Princeton University pressed a mold on a thin thermoplastic polymer film in 1995 and obtained a microstructure produced via hot embossing. As mentioned in their paper, there is a great need to develop low-cost technologies for the mass production of sub-50 nm structures, since such technology can bring enormous impacts to many areas of engineering and science. Not only will the future of semiconductor integrated circuits be affected, but the commercialization of many innovative devices which are far superior to current devices also hinges on the possibility of such a technology. Scanning electron beam lithography has been demonstrated to have a 10 nm lithography resolution, but using this technique for the mass production of sub-50 nm structures seems economically impractical due to the inherent low throughput of serial processing tools. Since its inception, this technology has developed extensively. Hot embossing is a high-throughput, low-cost manufacturing technology for microstructures on thermoplastic material surfaces (usually polymers) and is widely used in the fields of optics, biology, bionics, medicine and MEMS manufacturing [2]. In academia and industry, the roll-to-roll hot embossing of a microstructure using a metal micro-mold is considered to be the most reliable high-precision, high-efficiency and low-cost micro-/nanostructure manufacturing method, but the mold is a significant limitation for the development of hot embossing technology.

At present, the following methods are used in the manufacturing of hot-embossing molds: the direct construction method, the LIGA-based method and the intermediate method. Among them, the direct construction method for the manufacturing of molds includes micromachining, micro-EDM, laser processing and other aspects. Liu et al. [3] transferred a figure on silicon nanotubes to a PMMA plate. They then used the PMMA mold to emboss a polyethylene terephthalate (PET) nanochannel and used UV–lithography techniques to fabricate silicon nanomolds with a sub-micrometer width for the initial mold and PMMA as the intermediate body. Finally, they copied the microstructure to PET. Massiullo et al. [4] used a PFPE intermediate mold; the microstructures were copied from the silicon mold for the final thermoplastic material. The use of polymers to create the intermediate mold can improve the device yield of silicon nanomold components, reducing the cost of mold manufacturing. At the same time, the mold is limited by the characteristics of the polymer, which can lead to low mold accuracy and poor thermal stability. Kim and LOH [5] used single-crystal diamond tools to process continuous V-grooves and pyramidal microstructures onto a nickel-plated mold steel and nickel alloy. They studied the feasibility of using two-dimensional vibration-assisted cutting or elliptical vibration cutting to directly process the micro-pattern of a micro-mold, improving the machining quality of the microstructure. However, the tools used are easily damaged, and the cost is high. Ali et al. [6] used EDM to create a sub-millimeter-level microstructure which can be used as hot-embossing mold and faithfully copied the microstructure on the mold to the polymer surface; the authors also proposed that for the mass production of complex microstructures, an effective temperature change process and cooling system should be applied, meaning that the processing equipment is relatively complex. Rajab et al. [7] processed a microstructure/nanostructure on the surface of a metal mold with an ultra-short picosecond laser, forming a hydrophobic surface with a contact angle of 130°, and then transferred the microstructure to plastic via soft replication. Tran et al. [8] used silicon master plates to manufacture aluminum alloy (AA6061-T6) molds through hot pressing and studied the directional effect of the channels on AA6061-T6 molds produced via hot pressing based on a set of channels produced on the aluminum alloy. The authors found that aluminum alloy molds have a high quality and strength but low accuracy. The above-mentioned studies indicate that compared to the laser-induced superplasticity method, traditional metal-mold-manufacturing methods have lower safety levels, higher equipment requirements, more complex process procedures, higher overall costs, and lower production efficiency.

We utilized laser shock imprinting (LSI) technology to fabricate metal molds. In addition to copper, we also attempted to use materials with different hardnesses and yield strengths, such as aluminum and nickel, as mold materials. However, our research found that compared to copper, nickel required a larger number of pulses and had the most complex forming process. Aluminum exhibited a weaker flowability and performance compared to copper. Hence, we chose copper as the base material. The microstructures produced via laser-induced shock formation could meet the demands of various fields. In 2014, Shen et al. [9], from the University of Iowa, fabricated circular-arrayed microstructures with a diameter of 130 μm on 30 μm thick titanium foils. They found that the formed parts exhibited significantly improved mechanical properties, such as hardness and corrosion resistance. The cell culture experiments demonstrated that titanium foils with arrayed microstructures were more suitable for cell growth and proliferation than untreated titanium foils, a finding which is of great significance for biological cultivation. Furthermore, laser-induced shock imprinting technology can also be applied in the manufacture of molds for the hot embossing of LCP, PI, and PMMA materials.

The laser-imprinting method was reported as early as 20 years ago. In 2003, Xia et al. [10] proposed and demonstrated a nanopatterning technique, known as laser-assisted nanoimprint lithography (LAN), in which the polymer was melted with a single excimer laser pulse and then imprinted with a mold formed of fused quartz. LAN has been used to pattern nanostructures in various polymer films on a Si or quartz substrate with high fidelity over the entire mold area. The research findings revealed that LAN not only greatly shortened the imprint processing time but also significantly reduced the heating and expansion of the substrate and mold, leading to a better overlay alignment between the two. In 2006, Xia et al. [11] used a real-time imprint monitoring system (RIMS) and measured the processing time for laser-assisted nanoimprint lithography to be approximately 200 ns. They found that during this short period of time, the mold was fully pressed into the resist, resulting in a full pattern transfer with high fidelity. Their results also demonstrated the capability of RIMS for monitoring an ultra-fast imprint process.

Traditional micro-forming devices are relatively internally complex, with significant issues such as wear and repair during production, installation, and operation. For some metal materials with a poor forming ability, it is difficult to meet the requirements for manufacturing accuracy. Therefore, traditional micro-forming technology still has limitations. In order to solve the many problems with the current microfabrication technology and meet the market demand for large-scale production, according to the requirements for the mold in the process of roll-to-roll hot embossing, this paper presents a method for manufacturing high-precision metal micro-molds using LSI. LSI technology has the characteristics of non-contact, high precision, a small volume, material diversity, etc. The laser-induced shock wave can be used to accurately bend and shape the engineering parts. We previously conducted experiments to flatten and bend ultra-thin copper foil using laser shock flattening [12]. In the process of LSI, a nanosecond laser is usually used to generate a high-pressure shock wave, the shock wave acts on the material, changes the curvature of the sample, and introduces beneficial compressive residual stress, and the residual stress can strengthen the formed target [13,14]. Compared with the polymer-made hot-embossing mold, the metal mold has a higher strength and higher thermal stability. Compared with traditional metal-mold-manufacturing methods, the LSI method used in this study to produce a hot-embossing mold has the advantages of low cost and high efficiency. In this study, the microstructure of the initial mold was completely transferred to the copper foil as a metal intermediate and then transferred to the polymer surface via hot embossing. The formation mechanism and strengthening mechanism of copper foils with different thicknesses are discussed to verify the performance stability of the metal mold in the process of hot embossing.

## 2. Experiment and Characterization

### 2.1. LSI of Metal Micro-Mold

In Figure 1, from the top to the bottom are the confinement layer (K9 glass with a thickness of 5 mm), the ablative layer (black aluminum foil with a thickness of 45 μm), the copper foil and the initial mold (the mold material is nickel mesh, with a depth of 18 µm).

In the LSI experiment, the infrared laser (model Hercules-1000-TH, wavelength 1064 nm, pulse duration 7 ns) was selected as the irradiation source. The energy of the single pulse laser was 875 mJ, and the spot diameter was 5 mm. The pulse laser emitted from the nanosecond laser irradiated the ablative layer, the ablative layer absorbed the energy, which vaporized rapidly, and the generated plasma shock wave propagated down to the copper foil. When the pressure exceeded the dynamic yield limit of the copper foil, plastic deformation was induced, thus copying the shape of the initial mold to the copper foil. During the experiment, the confinement layer, the ablative layer, the copper foil and the initial mold are pressed tightly to reduce the loss of shock pressure in the transmission process.

Figure 2 shows the initial molds with different microstructure shapes used in this experiment, where the mold is a periodic array of the microstructure.

Figure 2a is a square hole array with a side length of 90 μm, Figure 2b is a round hole array with a diameter of 90 μm, and Figure 2c is a grating array with a width of 90 μm.

### 2.2. Hot Embossing of Flexible Substrate (PET)

As shown in Figure 3, in the hot-embossing experiment, the copper mold is tightly combined with polyethylene terephthalate (PET), and the experimental device for the upper and lower heating devices, the copper mold and PET are heated and pressurized with the device.

This experiment verified that the copper mold produced via LSI has excellent characteristics in terms of strength and thermal stability and provides a new concept for the manufacturing of polymer hot-embossing molds. In this experiment, PET was selected as the experimental object. PET has excellent physical and mechanical properties in a wide temperature range. Its glass transition temperature (Tg) is low, at approximately 80 °C, and the heating and cooling times can be reduced in the experimental process [15]. The hot-embossing experiment adopts the method of heating the mold and polymer substrate at the same time [16], and the selected experimental temperature is 90 °C, which is kept at approximately 10 °C above the Tg of PET. It was experimentally determined that the pressure for forming the microstructure on PET is 0.2–0.4 MPa. The polymer was heated above the Tg and kept at this temperature and pressure so that the polymer could flow to fill in the micro-cavity in the copper mold. Then, we removed the mold and polymer, cooled them to room temperature, and performed demolding after the polymer hardened.

The copper mold produced via LSI was used for repeated hot-embossing experiments. The profile curve and nano-hardness of the copper mold were measured before and after the experiment, and the changes were compared to verify the strength and thermal stability of the copper mold.

### 2.3. Measurements and Characterization

The surface morphology of the samples produced in the test was observed using a scanning electron microscope (SEM) (quanta250, ELECMI FEI, Hillsboro, OR, USA), observing the initial molds and copper molds with different microstructure shapes and the formed PET. The surface profile was observed and measured using an optical profiler (DVM5000, Leica, Wetzlar, Germany). The formed copper mold before and after several rounds of hot embossing and the formed PET were measured, and the changes in the microstructure-forming depths and profiles were compared. The mechanical properties were determined using a nano-hardness tester (Agilent U9820A, Nano Indenter G200, Accexp, Changsha, China, HVSA-1000A) with a 50 g load and a 15 s duration time to measure the nano-hardness and elastic modulus of the copper foil. The copper molds before and after the hot embossing experiment and their final results were compared. Then, the microstructures of the copper foils were characterized using TEM (JEM2100, JEOL Ltd., Tokyo, Japan). In order to reduce the error, each sample was measured at three different positions, and the average of the three measured values was used to represent the final measurement results.

### 2.4. Measurements and Characterization

In order to study the forming mechanism of the copper foil during LSI, we used ABAQUS software to numerically simulate the shock process. Through the simulation results, we could observe the process of the copper foil’s filling flow to the initial mold and the residual stress of the copper foil after LSI. The experiment involved the formation of materials at an ultra-high strain rate on a micro-scale; thus, the Johnson–Cook (J-C) model was selected. The assumptions on which the model was based are as follows: (1) as the copper foil used in the experiment is annealed copper foil, which is smaller than the hardness of the initial mold, the initial mold is assumed to be a rigid body; (2) as the diameter of the laser spot is much greater than the thickness of the ablative layer, it is assumed that the laser irradiation is uniform; (3) as the size of the copper foil in the horizontal direction is much greater than its thickness, the stress state of the copper foil can be regarded as one-dimensional, and the propagation of the laser shock wave in the copper foil can also be regarded as a one-dimensional strain wave [17]; (4) under the action of the laser shock wave, the deformation of the copper foil is isotropic [18]; (5) the flow stress and elastic properties include temperature dependence; and (6) it is assumed that there is no friction between the initial mold and the copper foil. The action time of a single laser pulse is 7 ns, while under the action of a confinement layer, the action time of the laser shock wave will increase. It is generally recognized that the action time of a shock wave is three times the pulse time because of the action of the confinement layer; hence, the action time of the force is 21 ns [19]. In 21 ns, the pressure produced by the shock wave changes with time. In Figure 4, P_max_ is the peak pressure of the shock wave, and t_p_ is the pulse duration [20]. Based on an interpolation algorithm, one can calculate the load at any time and introduce the corresponding load into the model.

## 3. Results

### 3.1. Manufacturing of a Metal Micro-Mold

In this experiment, copper foils of different thickness were used for LSI. The copper foil was annealed to eliminate its residual stress, rendering it is easier to form. After LSI, the processed part was strengthened; this step greatly increases the strength and is beneficial for repeated use as a mold in hot embossing. In this way, the microstructure of the processed copper mold can be retained in the subsequent hot-embossing process without deformation.

In the experiment, 20 μm, 50 μm, 80 μm and 100 μm copper foils were used to carry out LSI experiments. For samples of different thicknesses, the same initial mold and different laser shock parameters were selected. With the increase in the thickness of the copper foil, we increased the number of laser pulses in order to reach the maximum forming depth. The formed copper foil samples are shown in Figure 5.

With the increase in the thickness of the copper foil, the forming depth gradually decreased. Compared with the 20 μm copper foil, the difference from the 50 μm and 80 μm copper foil is minimal, and the difference from the 100 μm copper foil is highly apparent. From the profile curve, we can see that the forming depth of the 20 μm copper foil is the greatest, reaching 11.512 μm, while that of the 50 μm copper foil is 10,645 μm, and that of 80 μm copper foil is slightly lower, reaching 8.743 μm. For the 100 μm copper foil, the large number of laser pulses has no effect on the greater deformation of the copper foil. From Figure 5h, it can be seen that the forming depth is only 3.370 μm, which is far less than the values for the first three copper foils. Moreover, the surface morphology of the profile of the 100 μm copper foil indicates that its forming effect is poor, the forming depth and the surface of the microstructure are uneven, and the edge of the profile shows obvious defects.

As shown in Figure 6a, for the square-hole initial mold, the 20 μm thick copper foil was selected for LSI.

Because the pressure of a plasma shock wave exceeds the dynamic yield limit of a copper foil, the shocked copper foil flows, thus filling the holes in the initial mold. The back of the copper foil will also be deformed due to flow filling on the front side. For the 50 μm copper foil, as shown in Figure 6b, the forming process is similar to that of the 20 μm copper foil, but the 50 μm copper foil is thicker, so that the deformation on the back side is smaller than that on the 20 μm copper foil. As shown in Figure 6c, when the 80 μm copper foil was selected for the LSI experiment, in order to obtain a better microstructure-forming effect, more laser pulses were needed. The shock wave was transmitted through the copper foil, and flow occurred in each part of the shocked area; however, because the copper foil was thick, the flow on the inner and front surfaces was enough to fill in the holes in the initial mold, so that the back side could be kept flat. From the SEM images in Figure 6, it can be seen that the back side of the 20 μm copper foil has obvious deformation, while the back side of the 80 μm copper foil has no deformation. As shown in Figure 6d, for the 100 μm copper foil, the shock wave first propagated through the copper foil. In such a case, there will be attenuation when the thickness of the copper foil exceeds the forming limit of the shock wave; the thicker the copper foil is, the worse the micro-structure-forming accuracy is. Therefore, in the process of LSI, although there is no deformation on the back side of the 100 μm copper foil, the side in contact with the initial mold has insufficient liquidity, and the forming depth and forming accuracy are greatly reduced. From the experimental results, it can be concluded that in LSI experiments, in order to obtain complete imprinting shapes, with the increase in the target thickness, the number of laser pulses required must also be increased. If the thickness of the target is small, this will easily lead to formation and completely filled holes in the initial mold during LSI, but because of its small thickness, there will be deformation on the back side; if the thickness of the target is large, the increase in the number of laser pulses will not produce obvious deformation.

Figure 7 shows the overall distribution of stress in the 20 μm copper foil and 80 μm copper foil throughout the whole process of LSI.

Figure 7a shows the stress of the 20 μm copper foil under the action of the 1st laser pulse, the 3rd laser pulse and the 5th laser pulse in the LSI experiments; Figure 7b shows the stress of the 80 μm copper foil under the action of the 1st laser pulse, the 15th laser pulse and the 25th laser pulse; and Figure 7a,b shows the stress of the 20 μm copper foil and 80 μm copper foils. Based on the aspect of stress distribution, the process can be divided into two stages. (1) In the copper foil compression stage, after the first laser pulse, although the copper foil is not significantly deformed, a large stress is generated within it. (2) In the copper mold formation stage, as the number of laser pulses increases, the copper foil is gradually formed. With the change in the forming depth, different stress distributions appear in the copper foil, and the low-stress areas gradually disappear. As shown in Figure 7a, due to the high deformation of the back side of the copper foil, stress concentration occurs on the back surface. Figure 7b demonstrates that because the copper foil is thick, the back side is not deformed due to superplastic flow; hence, the stress in the overall shock area of the copper foil is more balanced.

As shown in Figure 8, the 20 μm and 80 μm copper molds were used in the hot-embossing experiment.

For the 20 μm copper mold, because deformation occurs on the back, during the repeated pressurization process, the convex microstructure on the front side will deform due to the lack of support; thus, the strength is low. However, the back of the 80 μm copper mold is flat, and there is no longitudinal deformation under the conditions of the hot-embossing experiment. Therefore, the 80 μm copper mold was finally selected for the hot-embossing experiment.

In order to transfer the microstructure to the surface of the polymer, the metal micro-molds formed via LSI were used in the hot-embossing experiment instead of the initial mold. In order to meet the diversity demand of hot-pressed products, three different microstructure-shaped molds were fabricated through large-scale integrated circuit experiments. Figure 9 shows a square microstructure mold, Figure 10a depicts a circular-hole microstructure copper mold labeled as 296, and Figure 10b represents a grating microstructure mold. In the SEM images, it can be seen that the three types of copper molds have a high forming accuracy.

### 3.2. Forming Mechanism of a Copper Mold

In order to verify the performance stability of the copper mold under the conditions of hot embossing, an optical profiler was used to measure the forming depth of the copper mold before and after the hot-embossing experiment. The forming depth of PET was also measured. The copper mold was subjected to five hot-embossing experiments. The measurement results are shown in Figure 11.

Figure 11a shows the comparison curve of the copper mold and the formed PET. Figure 11b–d, moving from top to bottom, shows the profile curve of the copper mold before the hot-embossing experiment, the copper mold after the hot-embossing experiment and the formed PET, respectively. It can be seen from Figure 11b that the forming depth of the microstructure of the copper mold before hot embossing is 5.554 μm, while in Figure 11c, the forming depth of the microstructure after hot embossing is 5.5095 μm, and the rebound level is approximately 0.8%. Therefore, it is proved that the copper mold processed via LSI has a high strength and can withstand the temperature and pressure conditions of hot embossing, as well as the external force perpendicular to the mold microstructure during the demolding process, so as to keep its shape and structural size unchanged. In Figure 11d, the surface microstructure depth of the formed PET is 5.439 μm, which shows that the copper mold formed via LSI can ensure the free flow and full filling of the PET in the hot-embossing experiment.

In order to observe the mechanical properties of the copper mold in the hot-embossing experimental process, the load–displacement curves of the annealed copper foil and the copper molds before and after the hot-embossing experiment were monitored.

As shown in Figure 12a, at the same indentation depth, the required indentation load of the annealed copper foil is the smallest, and the required indentation loads of the copper molds before and after the hot-embossing experiment are very close, being larger the value of the annealed copper foil.

Therefore, it can be proved that the mechanical properties of the formed parts are strengthened. Figure 12b shows the elastic modulus and nano-hardness values of the copper mold before and after hot embossing and the annealed copper foil. The elastic modulus and nano-hardness of the copper mold before and after hot embossing are improved compared with the annealed copper foil. After LSI, the elastic modulus of the copper mold before hot embossing increased from 70.5 to 91.3 GPa, which is an increase of 29.5%. The nano-hardness of the copper mold before hot embossing increased from 1.11 to 1.46 GPa, which is an increase of 24.0%. After the hot-embossing experiment, compared with the mold before hot embossing, the elastic modulus decreased to 88 GPa, a decrease of 3.61%, and the nano-hardness decreased to 1.32 GPa, a decrease of 9.58%, but its mechanical properties were still better than those of the annealed copper foil. The improvements in the mechanical properties of the copper mold can mainly be attributed to work hardening from the plastic deformation caused by the laser shock.

Laser shock processing is an effective method for achieving the deformation of metal materials at an ultra-high strain rate. The strain rate is usually as high as 106–107s-1, which is different from that of the traditional processing method [21]. The shock wave generated by the laser shock can cause the metal to rapidly undergo high plastic deformation, increasing the number of crystal defects in the metal and enhancing its mechanical properties [22]. On this basis, according to the evolution of the inner structure of the copper foil, the strengthening mechanism of the copper foil was analyzed in an LSI experiment.

In the experiment, the copper foil was annealed to reduce its crystal defects and residual stress. As shown in Figure 13, a small number of dislocation lines and dislocation tangles can be seen in the TEM image of the annealed copper foil.

The annealed copper foil before laser shock only has a small number of dislocation structures. The copper foil after the laser shock is shown in Figure 14. A large amount of dislocation cells (DCs), dislocation walls (DWs), dislocation tangles (DTs) and dislocation lines (DLs) are formed in the coarse grains, and these dislocation structures greatly increase the density of the crystal defects.

In the LSI process, the laser acted on the ablative layer to generate a shock wave with an ultra-high peak pressure, which could trigger most of the dislocation sources in the coarse grains in the beginning of the deformation process and caused the rapid plastic deformation of the copper foil. As the plastic deformation increased, the dislocation density increased, and then, the ultra-high peak pressure caused the DCs formed through plastic deformation to crack and multiply, as shown in Figure 14b,c.

There are two modes of plastic deformation of copper: dislocation slip and deformation twinning. In this experiment, the dislocation structure of the copper increased significantly, and no mechanical crystal caused by plastic deformation was observed. Therefore, dislocation slip is the main deformation mode in the LSI process.

In our previous experiments, the stability of LSI samples at high temperature was verified [23]. Although the thickness of the copper foil, with respect to the initial mold depth, is larger in this paper, the strengthening mechanism is the same as that in the previous experiment. Therefore, after the hot-embossing experiment, the mechanical properties of the copper mold changed only slightly, which proved that the copper mold formed via LSI can withstand the high temperature and pressure under hot-embossing conditions.

As shown in Figure 15, in the LSI experiment, the copper foil initially underwent tensile deformation to fill in the holes in the initial mold. After making contact with the bottom of the initial mold, the copper foil was extruded and deformed under the joint action of the laser pulse and the mold.

Another reason for the nano-hardness difference in the microstructure is the strengthening effect of LSI and contact stress. In the plastic deformation stage, the laser shock renders the material at the bottom of the microstructure subject to longitudinal pressure. At the same time, the contact between the copper foil and the bottom of the initial mold caused stress in the microstructure. This deformation process is similar to that in the research conducted by Shen et al. In their experiment, the ratio of the target thickness to the initial mold depth is 0.21 [24]. The residual stress field formed on the target surface, along with crystal defects such as grain refinement, dislocation and twin (as discussed in the literature [12]), can effectively improve the comprehensive mechanical properties of the material; thus, the nano-hardness will be significantly improved compared with the annealed copper foil. In the hot-embossing experiment, the pressure and temperature are not sufficient to cause changes in the stress and crystal defects in the microstructure of the copper mold; hence, the mechanical properties remain unchanged.

## 4. Conclusions

In this paper, LSI experiments were carried out on copper foil to manufacture metal micro-molds which could be used for hot-embossing experiments. In the LSI experiment, the profile curve, forming depth and surface morphology of the formed microstructure, as well as the numerical simulation carried out to explore the forming mechanism, showed that the forming depths of copper foils of different thicknesses were not different, but the back of the 80 μm copper foil was flatter, and there was no deformation as serious as that observed on the back of the 20 μm copper foil. Thus, the 80 μm copper foil was selected as the target material for the LSI experiment. The microstructure formed on the 80 μm copper foil was used as a metal micro-mold for the hot-embossing experiment to explore the integrity and fidelity of the PET surface microstructure in the hot-embossing experiment and to study the performance stability of the copper molds under the temperature and pressure conditions of hot embossing. The results showed that the copper mold produced in the LSI experiment meets the requirements of hot-embossing molds. This process provides a new concept for the manufacture of roll-to-roll hot-embossing molds and can promote the development of hot-embossing technology.

## Figures and Tables

**Figure 1 materials-16-05079-f001:**
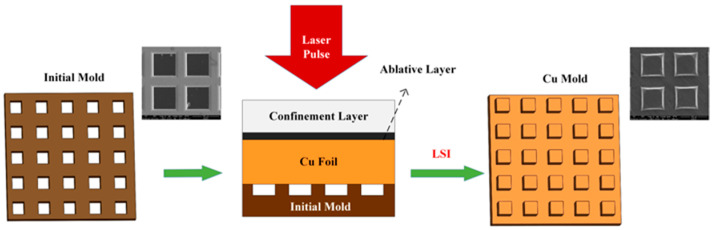
LSI for metal micro-mold.

**Figure 2 materials-16-05079-f002:**
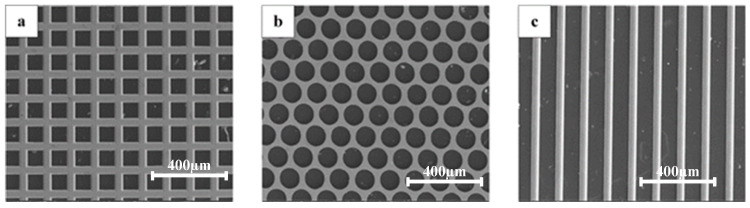
The SEM images of the initial molds with different microstructure shapes: (**a**) square hole, (**b**) round hole, (**c**) grating.

**Figure 3 materials-16-05079-f003:**
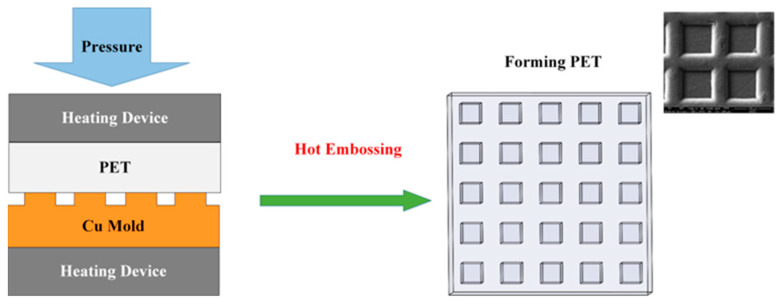
Hot embossing of flexible substrate (PET).

**Figure 4 materials-16-05079-f004:**
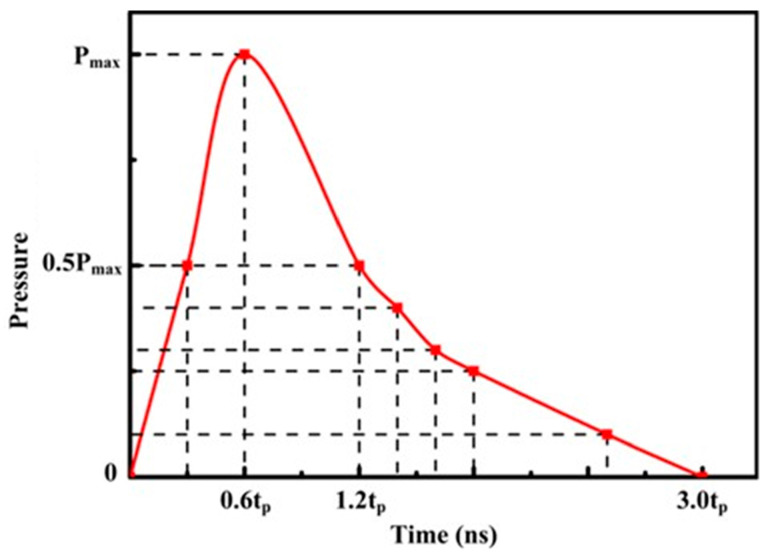
Pressure curve of the shock wave over time.

**Figure 5 materials-16-05079-f005:**
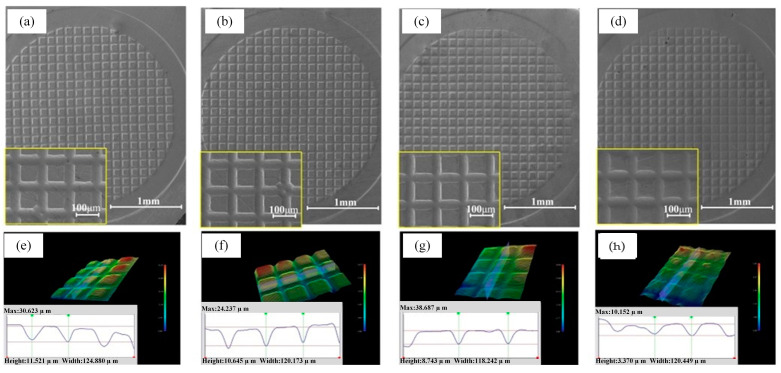
SEM images and profile curves of microstructures on copper foils of different thicknesses: (**a**,**e**) 20 μm copper foil, (**b**,**f**) 50 μm copper foil, (**c**,**g**) 80 μm copper foil, (**d**,**h**) 100 μm copper foil.

**Figure 6 materials-16-05079-f006:**
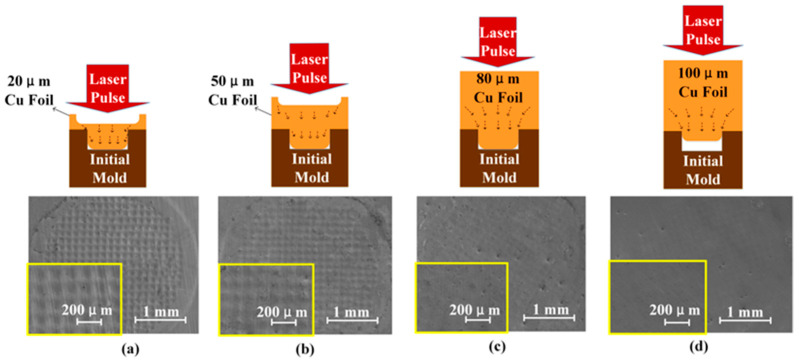
Images of copper foil back sides and schematic diagram of the forming process of copper foils with different thickness: (**a**) 20 μm copper foil, (**b**) 50 μm copper foil (**c**) 80 μm copper foil, (**d**) 100 μm copper foil.

**Figure 7 materials-16-05079-f007:**
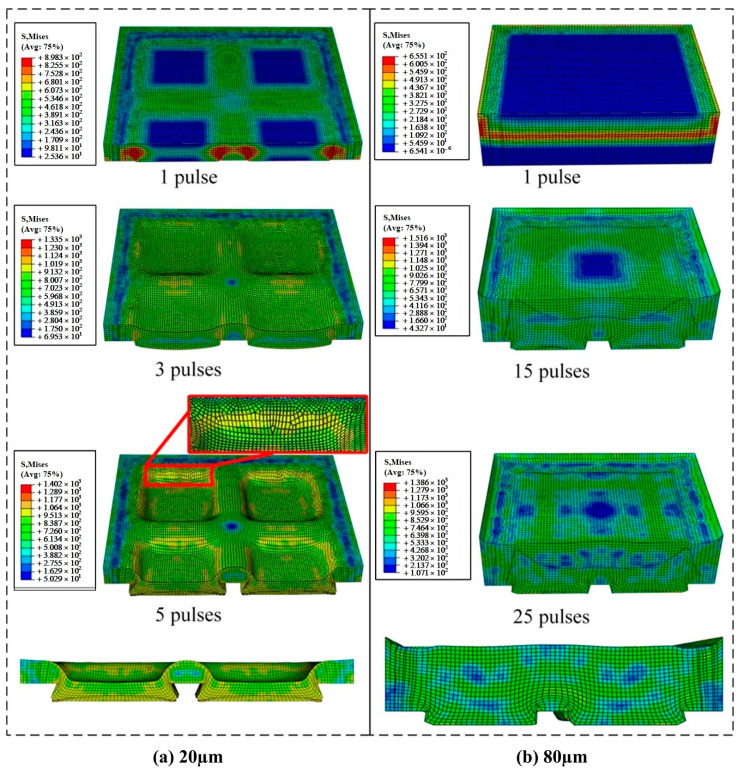
Deformation process of the shock imprinting of 20 μm (**a**) and 80 μm (**b**) copper foils.

**Figure 8 materials-16-05079-f008:**
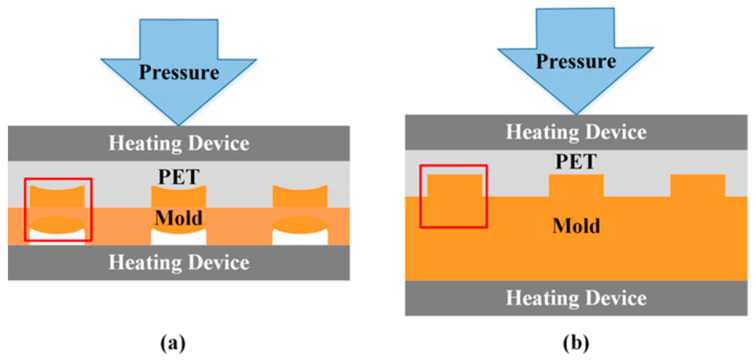
Comparison of mold strength: (**a**) 20 μm copper mold, (**b**) 80 μm copper mold.

**Figure 9 materials-16-05079-f009:**
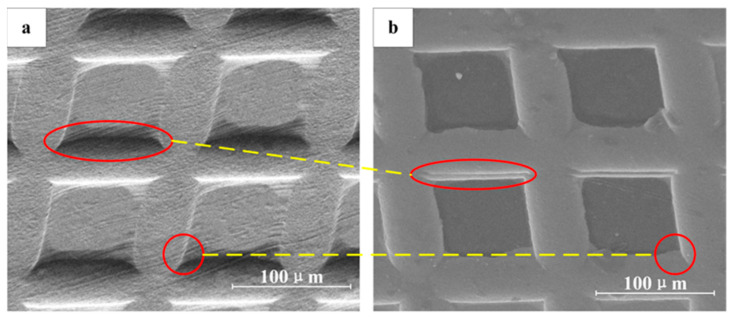
(**a**) SEM images of microstructures on copper molds. (**b**) SEM images of formed microstructures on PET.

**Figure 10 materials-16-05079-f010:**
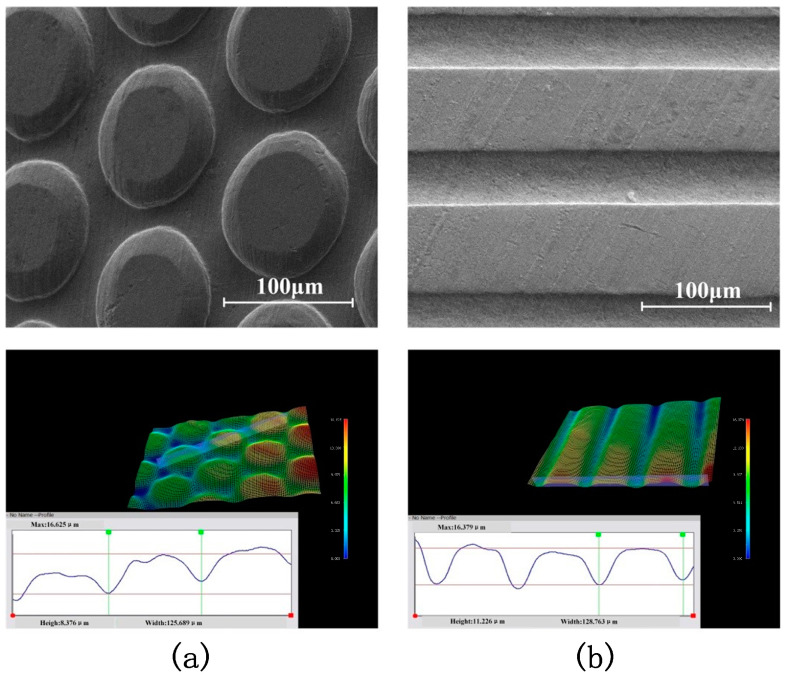
Copper molds with different microstructure shapes: (**a**) round-hole copper mold, (**b**) grating copper mold.

**Figure 11 materials-16-05079-f011:**
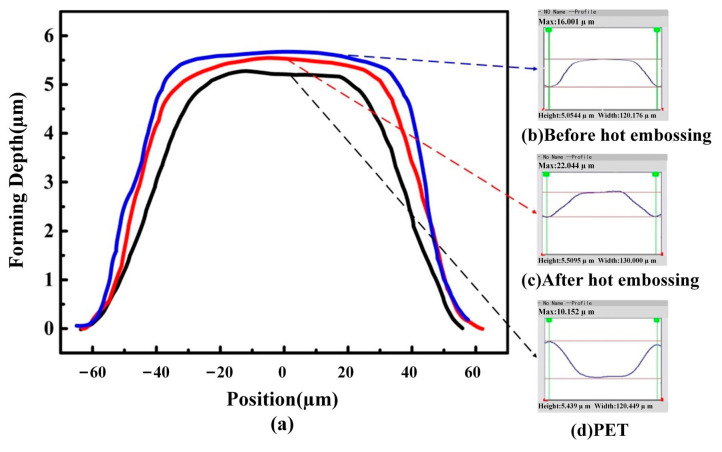
Comparison of microstructure-forming depths between the PET and copper mold before and after hot embossing: (**a**) comparison curve, (**b**) forming depth of copper mold before hot stamping, (**c**) forming depth of copper mold after hot stamping, (**d**) forming depth of PET.

**Figure 12 materials-16-05079-f012:**
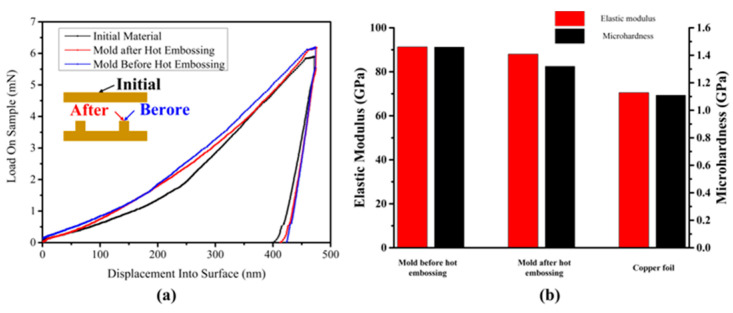
Nano-indentation results of a copper mold before and after hot embossing and annealed copper foil. (**a**) Load–displacement curves. (**b**) Elastic modulus and nano-hardness.

**Figure 13 materials-16-05079-f013:**
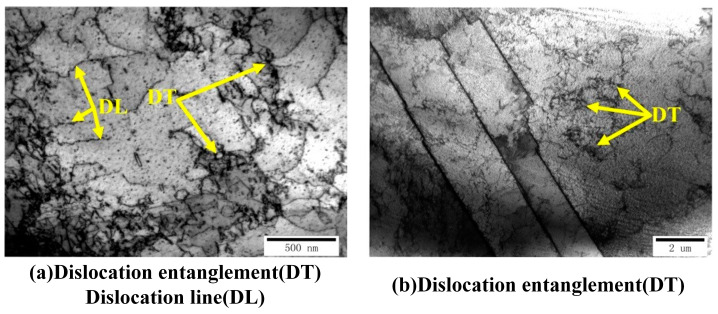
TEM images of the dislocation structure in the annealed copper foil.

**Figure 14 materials-16-05079-f014:**
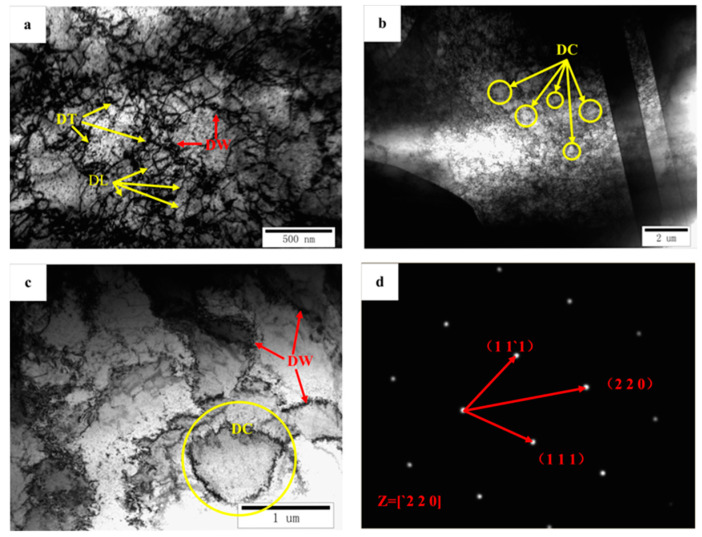
TEM images of copper foil after laser shock: (**a**) DTs, DLs, and DWs; (**b**) DCs and dislocations; (**c**) DCs and DWs; (**d**) selected area electron diffraction (SAED) pattern of copper foil.

**Figure 15 materials-16-05079-f015:**
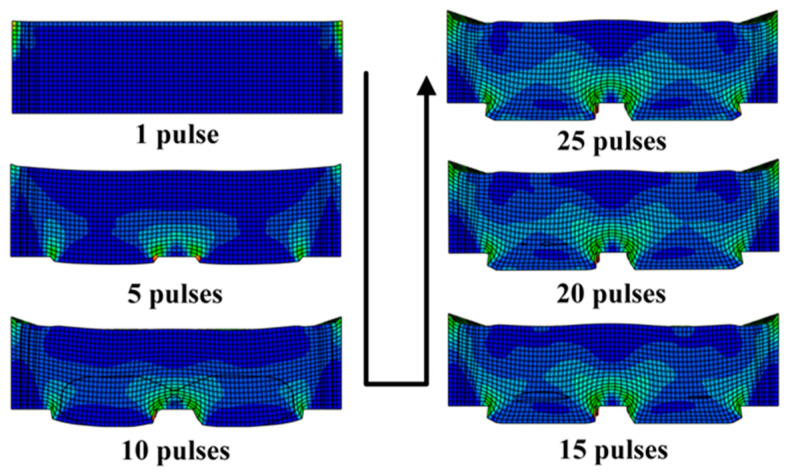
The strain change in the copper mold with the number of laser pulses.

## Data Availability

The data presented in this study are available on request from the corresponding author.

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
