# Peer review of "Fabrication of a Hot-Embossing Metal Micro-Mold through Laser Shock Imprinting"

_materials, 2023, doi:10.3390/ma16145079_

Round 1
Reviewer 1 Report
This manuscript described a new way of fabricating metal mold for hot embossing. It is indeed difficult to make metal mold, much more difficult than making silicon mold. Thus the paper is valuable.
My main concern is about the hot embossing (nanoimprint) experiment. The author mentioned that PET has Tg of 80 deg, so they used 90 deg to do the imprint. As a result, the imprinted pattern depth is less than the mold pattern depth. There is no reason that the author cannot increase the temperature. For PMMA having Tg of 105 deg, people usually do imprint at around 150 deg. If the authors increase the imprint temperature to, for example, 130 deg, I believe the imprint into PET will be quite easy. Hot embossing of thermoplastic polymers has been so well studied in the past three decades. The author may just try to imprint at high enough temperature, then simply present some imprint results to show the application of the metal mold. This way, the author can cut off 90% of the text regarding imprint results and its discussion (the manuscript is too long, so it is good to cut it short).
Below are some minor issues.
1. Page 1 line 44, carbon nanotube can be made by EUV lithography?
2. What is the material for the "initial mold"? Silicon? And what is the pattern depth for it?
3. P3 Line 96, 1064nm is not UV.
Figure 2 caption, missed the scale bar (the one inside image is hard to see).
Figure 4, the y-axis unit should be arbitrary unit, not GPa, since we don't know Pmax.
For the inserted AFM scanning profile, it is all difficult to see (too small or too blurred).
There are numerous mistakes, notably the sentence structures (the authors are encouraged to use short, simple, sentence structures), yet overall the writing is quite easy to understand.
Author Response
Dear Reviewer1:
Thank you for your valuable comments and feedback. We have revised our manuscript based on your suggestions. We hope that with this revision, the manuscript will meet the publication standards of Materials. Our point-by-point response to your comments is detailed in the following pages. Please consider our response and appreciate any further suggestions. We look forward to seeing our manuscript published in your journal. Thanks a lot.
Sincerely Yours,
Jingbin Hao
PS: Please see the attachment.

Reviewer 2 Report
Dear Authors,
This paper presents an interesting and detailed study on the fabrication of metal micro molds using LSI. The experimental methods are well explained and the results are clearly and appropriately presented. The conclusions are also supported by the results and are scientifically sound.
As suggestions, I would like to propose the following points for consideration:
-
Please provide more detailed information on the comparison between the method proposed in this paper and traditional metal mold fabrication methods. For example, a comparison in terms of cost and efficiency would make it easier for readers to understand.
-
Please provide more detailed explanation on whether the method proposed in this paper can be applied to other materials or other hot embossing applications.
These are my comments and suggestions. This paper presents interesting and valuable research, and I look forward to further developments in this field.
Sincerely,
Author Response
Dear Reviewer2:
We would like to express our sincere gratitude for your thoughtful review of our manuscript. Your comments and feedback have been incredibly valuable in improving the quality of our work. We hope that with this revision, the manuscript will meet the publication standards of Materials. Our point-by-point response to your comments is detailed in the following pages. Please consider our response and appreciate any further suggestions. We look forward to seeing our manuscript published in your journal. Thanks a lot.
Sincerely Yours,
Jingbin Hao
PS: Please see the attachment.

Reviewer 3 Report
This paper is valuable because it performed micropattern formation by laser heating and investigated the molding conditions. The results obtained are valid and useful for future reference. However, the laser imprinting method was originally reported 20 years ago, and it is necessary to specify it in this paper. The information of the paper and the reference paper is described below.
Ultrafast patterning of nanostructures in polymers using laser assisted nanoimprint lithography
Qiangfei Xia, Chris Keimel, Haixiong Ge, Zhaoning Yu, Wei Wu, Stephen Y. Chou
Journal: Applied Physics Letters
Appl. Phys. Lett. 83, 4417–4419 (2003)
DOI: https://doi.org/10.1063/1.1630162
In situ real time monitoring of nanosecond imprint process
Qiangfei Xia, Zhaoning Yu, He Gao, Stephen Y. Chou
Journal: Applied Physics Letters
Appl. Phys. Lett. 89, 073107 (2006)
DOI: https://doi.org/10.1063/1.2335952
Those who know NIL are familiar with this famous paper by Professor Chou of Princeton, so it is necessary to clearly explain what the micro order does and what its purpose is.
Therefore, please cite Professor Chou et al.'s original paper and clearly explain in the introduction what problems and needs there are in the micro order and what you are trying to do. Without this statement, the manuscript will be rejected.
English is okay, but there are sentencesthat are difficult to understand, so please check.
Author Response
Dear Reviewer3:
We want to extend our heartfelt appreciation for your insightful evaluation of our manuscript. Your review has been tremendously beneficial in enhancing the caliber of our work. We are confident that, with the revisions made, the manuscript now adheres to the exacting publication standards set by Materials. A comprehensive, point-by-point response to your comments can be found in the subsequent pages. We kindly request you to peruse our response and welcome any additional suggestions you may have. We eagerly anticipate the opportunity to have our manuscript published in your esteemed journal. Thank you immensely for your invaluable support.
Sincerely Yours,
Jingbin Hao
PS: Please see the attachment.

Round 2
Reviewer 3 Report
This manuscript is revised. I think this paper has a potential to publish.